# Anthropogenic combustion iron as a complex climate forcer

Hitoshi Matsui [1,2], Natalie M. Mahowald [2], Nobuhiro Moteki[3], Douglas S. Hamilton [2], Sho Ohata [3], Atsushi Yoshida[3], Makoto Koike[3], Rachel A. Scanza[4] & Mark G. Flanner [5]

Atmospheric iron affects the global carbon cycle by modulating ocean biogeochemistry through the deposition of soluble iron to the ocean. Iron emitted by anthropogenic (fossil fuel) combustion is a source of soluble iron that is currently considered less important than other soluble iron sources, such as mineral dust and biomass burning. Here we show that the atmospheric burden of anthropogenic combustion iron is 8 times greater than previous estimates by incorporating recent measurements of anthropogenic magnetite into a global aerosol model. This new estimation increases the total deposition flux of soluble iron to southern oceans (30–90 °S) by 52%, with a larger contribution of anthropogenic combustion iron than dust and biomass burning sources. The direct radiative forcing of anthropogenic magnetite is estimated to be 0.021 W m$^{-2}$ globally and 0.22 W m$^{-2}$ over East Asia. Our results demonstrate that anthropogenic combustion iron is a larger and more complex climate forcer than previously thought, and therefore plays a key role in the Earth system.

[1] Graduate School of Environmental Studies, Nagoya University, Nagoya, Japan 464-8601. [2] Department of Earth and Atmospheric Sciences, Cornell University, Ithaca, NY, USA 14853. [3] Department of Earth and Planetary Science, Graduate School of Science, University of Tokyo, Tokyo, Japan 113-0033. [4] Atmospheric Sciences and Global Change Division, Pacific Northwest National Laboratory, Richland, WA, USA 99352. [5] Climate and Space Sciences and Engineering, University of Michigan, Michigan, MI, USA 48109. Correspondence and requests for materials should be addressed to H.M. (email: matsui@nagoya-u.jp)

Iron is recognized as an essential micronutrient for ocean primary productivity and modulates marine ecosystems, global carbon cycle, and atmospheric carbon dioxide ($CO_2$) uptake[1–7]. Since atmospheric soluble iron deposited to the ocean is a major source of bioavailable iron to the remote surface ocean[8], understanding its magnitude, including the contribution from human activities, is important to quantify. The two main sources of soluble iron in the atmosphere are iron-containing mineral dust (Dust) and combustion iron which is mainly emitted from anthropogenic (AN; mostly from fossil fuel) combustion and biomass burning (BB)[5,9–13]. Compared to the iron contained in dust, combustion iron has a smaller emission flux but higher solubility[1,2,9]. As a result, combustion iron may contribute up to 50% of the soluble iron over remote ocean regions[1,2,14,15] (termed high-nutrient low-chlorophyll), where ocean primary productivity is iron-limited[3,16]. Soluble iron from Dust sources is currently considered more important than anthropogenic combustion iron (AN iron)[15]. However, current estimations of soluble iron deposition have large uncertainties in terms of both total fluxes and source contributions, due to a shortage of observational constraints.

In addition to the ocean biogeochemistry effect, combustion iron may have a heating effect in climate through absorption of solar radiation. Moteki et al.[17] recently found unexpectedly high atmospheric concentrations of magnetite, a strong light-absorbing iron oxide, through particle-resolved measurements of continental outflows from anthropogenic combustion sources in East Asia. The shortwave absorption by these magnetite particles was estimated in that study to be at least 4–7% of black carbon (BC) absorption[17].

Here, we incorporate the observed findings of Moteki et al.[17] into a global aerosol model[18,19] and show the following three results. First, we show that the atmospheric burden of AN iron should be eight times greater than previous estimates[1] to capture the observed magnetite concentrations, which also allows the model simulations to better match the previous measurements[1,2]. Second, we show that the direct radiative forcing (DRF) of AN magnetite may be comparable to that of brown carbon (BrC) through the first global evaluation of the heating effect of AN magnetite. Third, our new estimates, combining model simulations and iron observations, suggest that AN iron may dominate the total deposition flux of soluble iron and its variability in the Anthropocene, especially over southern oceans (mid-latitudes and high-latitudes), where the ocean biogeochemistry is likely to be iron limited.

## Results

**Global model simulations.** We conducted global aerosol model simulations for both present-day (PD) and preindustrial (PI)

conditions with two treatments of AN iron emissions (BASE and NEW; see Methods and Supplementary Table 1). In the BASE simulation, we used the emission flux of AN iron from Luo et al.[1] (0.66 Tg y$^{-1}$), which is consistent with other emission estimates of AN iron (0.51–0.87 Tg y$^{-1}$)[20,21]. In the NEW simulation, the emission flux of AN magnetite was estimated using the observed magnetite to BC mass ratio (0.40) in East Asia[17]. We assume the same spatial patterns of emissions for the BASE (for AN iron) and NEW (for AN magnetite) simulations. On the basis of the same set of measurements[17], a smaller aerosol emission size distribution is applied in the NEW simulation (for AN magnetite) than the BASE simulation (for AN iron) (see Methods). Other simulation settings (e.g., BB iron emission[1], online dust emission scheme[22–24]) are the same between the two simulations. Our global aerosol model was evaluated previously[19], except for magnetite, which was added to the model for this study.

In NEW simulation, the fraction of AN magnetite to AN iron (magnetite fraction) was assumed to be 40% in offline estimation of the total and the soluble iron concentrations and their deposition flux. This magnetite fraction has not been constrained by observations and might be much higher or lower than 40% depending on the combustion sources[25,26] (Supplementary Note 1). We define the uncertainty range of the magnetite fraction between 20% and 80%, and discuss the total and the soluble iron estimates within this uncertainty range (Supplementary Note 1).

**Emission flux and atmospheric burden.** The emission flux of AN magnetite in the NEW simulation (1.3 Tg y$^{-1}$) is double the AN iron in the BASE simulation (0.66 Tg y$^{-1}$) (in PD) (Fig. 1a). The AN magnetite in the NEW simulation has a factor of 1.5 longer lifetime (3.2 d) than AN iron in the BASE simulation (2.1 d) because AN magnetite in the NEW simulation has a smaller particle size (based on observations) and hence a slower deposition rate than AN iron in the BASE simulation. As a result, the atmospheric burden of AN magnetite in the NEW simulation is 3.1 times greater than that of AN iron in the BASE simulation (Fig. 1b). The concentrations of AN magnetite are highest over East Asia and Europe (Supplementary Fig. 1). The magnetite/BC mass ratio is greater than 0.50 in these regions and 0.30 on a global average (Supplementary Fig. 1d). The ratio decreases during atmospheric transport because magnetite has a larger particle size distribution and shorter lifetime than BC (4.5 d).

Compared to the observations, the simulated magnetite/BC mass ratio is underestimated by 80–85% in the BASE simulation, even under the assumption that all AN iron is magnetite (Supplementary Fig. 2). In NEW simulation, the magnetite/BC mass ratio agrees with the measurements within ±20% over the

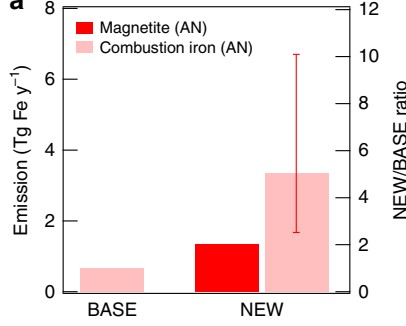
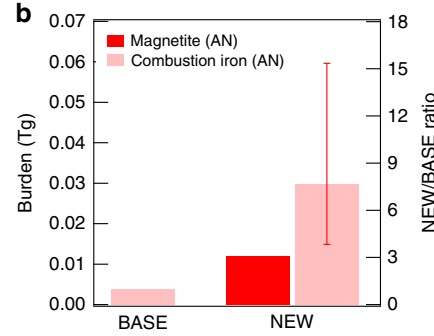

**Fig. 1** Budgets of magnetite and combustion iron. **a** Emission and **b** atmospheric burden of magnetite (red) and combustion iron (pink) from anthropogenic (AN) sources for the BASE and NEW simulations (global and 5-year mean). The fraction of anthropogenic magnetite to anthropogenic combustion iron was assumed to be 40% in the NEW simulation (see Supplementary Note 1). Error bars show the uncertainty range by the assumption of the fraction of magnetite to anthropogenic combustion iron (defined between 20% and 80%)

main continental outflow regions in East Asia (Supplementary Fig. 2a) and Europe (Supplementary Fig. 2b), where the simulated magnetite concentrations are the highest (the measurement at Zeppelin is not used for estimating magnetite emission used in the NEW simulation). This result suggests that the magnetite emission flux and its spatial pattern used in NEW simulation are better able to match the limited observations. Observations of magnetite concentrations (or magnetite/BC) are missing in North America and the Southern Hemisphere. In addition, emission sources of magnetite are currently not well known. However, the estimation of the possible emission sources of magnetite from BC emission inventories and the underestimation of combustion iron around Southern Africa and Australia support the validity of extending the magnetite results in East Asia to the Southern Hemisphere, where we focus on soluble iron estimates (Supplementary Note 2).

When the mass fraction of AN magnetite to AN iron is 40% (NEW simulation), the emission flux ($3.4 \, \text{Tg} \, \text{y}^{-1}$) and the burden (0.030 Tg) of AN iron are 5.0 and 7.7 times greater than those in the BASE simulation, respectively (Fig. 1). The emission flux and the burden of total combustion iron (AN+BB) in the NEW simulation are 2.5 and 2.8 times greater than those in the BASE simulation, respectively (Supplementary Fig. 3). The estimates of AN iron are doubled or halved within the assumed magnetite fraction ranging between 20% and 80% for the emission flux ($1.7–6.7 \, \text{Tg} \, \text{y}^{-1}$, 2.5–10 times greater than the BASE simulation) and the atmospheric burden (0.015–0.060 Tg, 3.8–15 times greater than the BASE simulation) (Fig. 1, Supplementary Fig. 3, and Supplementary Note 1).

Comparisons of the model simulations with the total iron (AN+BB+Dust) observations at Gosan in East Asia[1] show that the iron emissions used in the NEW simulation are more consistent with the measurements than those used in the BASE simulation and Luo et al.[1]. The NEW simulation (annual mean total iron concentrations are $0.66 \, \mu\text{g} \, \text{m}^{-3}$) has a good agreement with the total iron observations[1] ($0.63 \, \mu\text{g} \, \text{m}^{-3}$), whereas the simulations in the BASE case ($0.36 \, \mu\text{g} \, \text{m}^{-3}$) and Luo et al.[1] ($0.21 \, \mu\text{g} \, \text{m}^{-3}$) underestimate iron concentrations by a factor of 2 or 3 (Supplementary Fig. 4). Supplementary Fig. 4 also shows that the NEW simulation can simulate the observed iron concentrations in both summer and winter (when the contribution from Dust iron is limited, i.e., AN iron is dominant).

Comparisons of the simulated surface total iron concentrations with measurements[2] show that when we focus on data with an appreciable contribution from combustion iron (>20% of total in simulations), the total iron concentrations are underestimated in the BASE simulation by 45% (Supplementary Fig. 5a). Conversely, the NEW simulation has better agreement with measurements, with observed and calculated median iron concentrations almost identical (27 and $26 \, \text{ng} \, \text{m}^{-3}$, respectively) (Supplementary Fig. 5b). The total iron concentrations are underestimated by 73% and 7% in the BASE and NEW simulations, respectively, over the South Indian ocean (compared with observation data by Witt et al.[27]). Simulated total iron concentrations are $0.15 \, \text{ng} \, \text{m}^{-3}$ and $0.53 \, \text{ng} \, \text{m}^{-3}$ in the BASE and NEW simulations, respectively, over the Southern ocean around Tasmania, Australia (120°–140 °E and 45°–60 °S), and these concentrations are lower than the observed iron concentrations ($0.8 \, \text{ng} \, \text{m}^{-3}$) at Cape Grim in Tasmania[14] for the baseline marine air in the region. The NEW simulation also has better agreement with measurements for the deposition flux of total iron, especially in the Southern Hemisphere (Supplementary Fig. 6). The NEW simulation is therefore more realistic than the BASE simulation, in terms of surface concentrations and deposition flux of total iron, including around Southern Africa and Australia, and their outflow regions in the Southern Hemisphere, where ocean primary productivity is considered to be iron limited (Supplementary Fig. 7). Since combustion iron is the largest source and iron/BC ratios are underestimated around Southern Africa and Australia in our simulations (Supplementary Fig. 7), the combustion iron emissions (from AN or BB sources, or a combination of both) are likely underestimated within these regions (Supplementary Note 2). We therefore suggest that the underestimation of AN iron emissions is a main source of the underestimation of total iron around Southern Africa and Australia (Supplementary Note 2).

**Heating effect of magnetite.** Magnetite is likely the most efficient shortwave absorber among the iron-oxide minerals in the atmosphere, as the imaginary part of the refractive index for magnetite is similar to that for BC[28,29]. The global-mean DRF (defined as the difference between PD and PI, see Methods) of AN magnetite is $0.0051 \, \text{W} \, \text{m}^{-2}$ in the NEW simulation. In our simulations, the magnitude of AN magnetite DRF depends on the accuracy of BC DRF because we use the magnetite/BC ratio for estimating magnetite emissions. BC DRF in our simulations ($0.13 \, \text{W} \, \text{m}^{-2}$) is within the range of estimates by other global aerosol models ($0.05–0.37 \, \text{W} \, \text{m}^{-2}$ for fossil fuel and biofuel emissions)[30]. However, this BC DRF is 4–7 times smaller than BC DRF estimates constrained by surface solar radiation observations[31], likely due to an underestimate in emissions of BC. When we assume a similar underestimation of DRF for magnetite, the global-mean DRF of AN magnetite quadruples to $0.021 \, \text{W} \, \text{m}^{-2}$. The estimated global-mean DRF of AN magnetite ($0.0051–0.021 \, \text{W} \, \text{m}^{-2}$) is smaller than that of BrC ($0.016–0.038 \, \text{W} \, \text{m}^{-2}$), which was derived assuming the refractive index for moderately and strongly absorbing BrC[32] (see Methods), but these DRF values for magnetite and BrC are comparable within their uncertainty ranges (the maximum DRF by AN magnetite ($0.021 \, \text{W} \, \text{m}^{-2}$) vs. DRF by moderately absorbing BrC ($0.016 \, \text{W} \, \text{m}^{-2}$)). Since some recent studies[33–35] reported a lower imaginary part of the refractive index of BrC than the moderately absorbing BrC value used here (Supplementary Note 1), BrC could have a lower DRF if these refractive indices were used in our simulations.

AN magnetite DRF in the maximum estimation (scaled DRF considering the underestimation of BC DRF) is $0.22 \, \text{W} \, \text{m}^{-2}$ over East Asia (100°–140 °E and 20°–45 °N) (Fig. 2a) and is greater than BrC DRF (moderate absorption) over the most of the Northern Hemisphere (Fig. 2b). However, the uncertainties in magnetite (maximum vs. minimum estimates) and BrC (moderate vs. strong absorption) easily change the magnitude relationship of DRF between magnetite and BrC (Supplementary Fig. 8). Further observational constraints of the light-absorbing properties and their global distribution are needed for reducing the uncertainty ranges of their DRFs.

Magnetite/BC global-mean DRF ratio is 0.041 in the NEW simulation. The magnetite/BC DRF ratio is 0.05–0.30 near the source regions and the Northern Hemisphere and < 0.02 over the remote regions and the Southern Hemisphere (Supplementary Fig. 9). The Magnetite/BC DRF ratio over East Asia of 0.05–0.10 agrees well with the local magnetite/BC heating ratio derived from the in-situ aircraft data in this region[17]. Since BC DRF is small or negative over the U.S. and Europe, positive DRF of AN magnetite may be important over these regions. Estimates for snow radiative forcing of magnetite (see Methods) suggest it may also be greater than the BrC snow radiative forcing over the northeast part of China (Supplementary Fig. 10). Therefore, AN magnetite may have non-negligible impact on snow-darkening (and potentially on snow melting and water cycle) over this region.

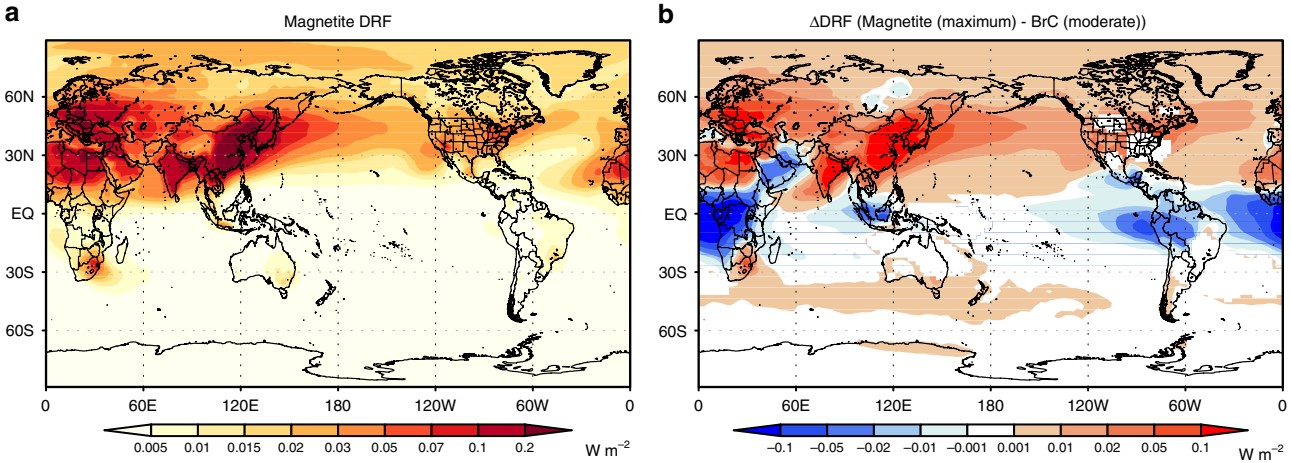

**Fig. 2** Spatial distributions of magnetite direct radiative forcing. **a** Direct radiative forcing (DRF) of anthropogenic magnetite (maximum estimate) in the NEW simulation (5-year mean). **b** DRF difference between anthropogenic magnetite (maximum estimation) and brown carbon (BrC) (moderately absorption) (5-year mean)

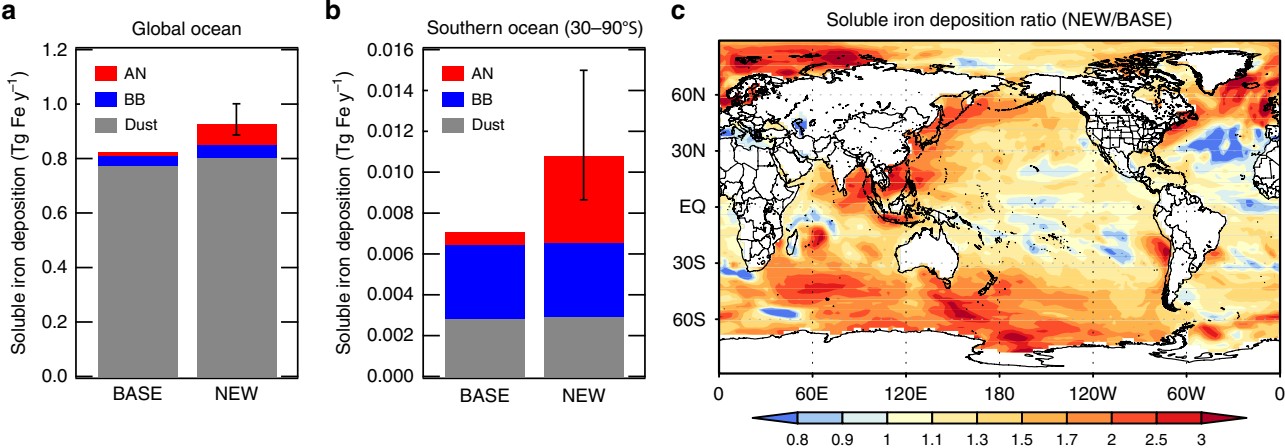

**Fig. 3** Soluble iron deposition flux to the ocean. **a**, **b** The deposition flux over **a** the global ocean and **b** the southern ocean at mid- and high-latitudes (30–90°S) from anthropogenic (AN, fossil fuel+biofuel), biomass burning (BB), and mineral dust (Dust) sources for the BASE and NEW simulations (5-year mean). Error bars show the uncertainty range by the assumption of the fraction of magnetite to anthropogenic combustion iron (between 20% and 80%). **c** Global distributions of the ratio of total soluble iron deposition flux (AN+BB+Dust) between the NEW and the BASE simulations (5-year mean)

**Impact on soluble iron estimates**. Total iron was simulated in our model (shown above). Soluble iron concentrations were then estimated from the product of total iron concentrations and the percentage solubility for each source (AN, BB, and Dust) (see Methods). Solubility was estimated from a global aerosol model which considers emissions of insoluble and soluble iron and the subsequent atmospheric formation processes of soluble iron explicitly[36]. The solubility of iron deposited to the global ocean is 7.7%, 11%, and 4.0% for AN, BB, and Dust sources, respectively, in the NEW simulation (Supplementary Figs. 11a-c). The solubility increases during atmospheric transport from the source regions to the remote oceans, which is consistent with some measurements[37,38]. The same spatial distribution of solubility is applied to the NEW and the BASE simulations, but solubility over the global ocean is slightly different (7.2%, 11%, and 4.1% for AN, BB, and Dust sources in the BASE simulation) because of different spatial distributions of iron between the two simulations. The deposition flux of soluble iron to the global ocean is 0.012, 0.039, and 0.77 Tg y$^{-1}$ for AN, BB, and Dust sources, respectively, in the BASE simulation (Fig. 3a). The soluble iron deposition fluxes for AN and BB sources (BASE simulation) reasonably agree

with previous estimates[15,36], whereas that for Dust sources is greater than previous studies by 50–150%[1,15,36]. This higher deposition flux of soluble iron from Dust sources is mainly due to higher emission flux of dust iron (95 Tg y$^{-1}$) than other studies (35–79 Tg y$^{-1}$)[1,15,21,36,39]. Therefore, our estimates shown below may overestimate the contribution from Dust sources, while underestimating the contributions from AN and BB sources, compared with other modeling studies.

In the NEW simulation, the deposition flux of total (AN+BB +Dust) soluble iron to the ocean increases by 12% globally (compared with the BASE simulation) and 52% over the mid- and high-latitudes in the Southern Hemisphere (30–90°S) (Fig. 3a, b). The total soluble iron deposition flux increases by more than 50% (the NEW/BASE ratio is greater than 1.5) over the southern oceans (30–90 °S), showing a large impact of our new estimates on soluble iron deposition over these regions (Fig. 3c). In contrast, the enhancement of the total soluble iron deposition flux is less than 20% (the NEW/BASE ratio is smaller than 1.2) over the Eastern Equatorial Pacific (Fig. 3c). The spatial pattern of the NEW/BASE ratio is consistent with the spatial distribution of the contribution of AN iron to the total deposition flux of soluble

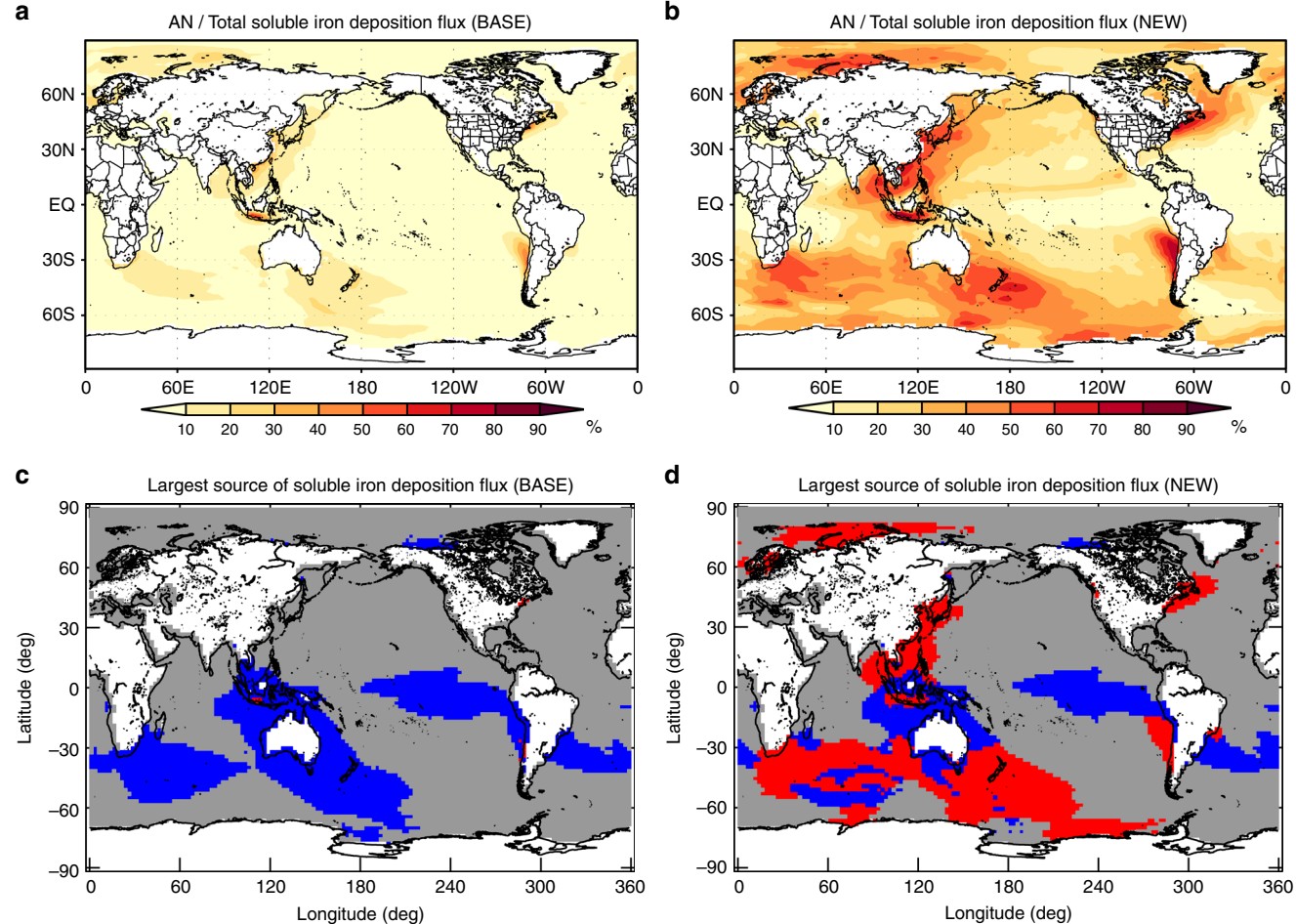

**Fig. 4** Source contributions of soluble iron deposition flux. **a**, **b** Global distributions of the contribution of anthropogenic (AN) sources to total soluble iron deposition flux (AN+BB+Dust) for **a** the BASE and **b** the NEW simulations (5-year mean). **c**, **d** Global distributions of the largest source of soluble iron deposition flux for **c** the BASE and **d** the NEW simulations. AN-, BB-, and Dust-dominant regions are shown by red, blue, and gray, respectively. Figures were made using IGOR Pro, WaveMetrics Inc., Oswego, OR

iron (Supplementary Figs. 11d-f). The contribution of AN iron to the total soluble iron deposition flux increases from 9.2% (BASE) to 39% (NEW) over the southern oceans (30–90 °S). The contribution from the AN source (39%) is greater than that from BB and Dust sources (34% and 27%, respectively) (Fig. 3b), suggesting that AN iron is the largest source of soluble iron deposition flux over the southern oceans (30–90 °S) in the NEW simulation.

In the BASE simulation, the contribution of AN iron to total soluble iron deposition flux is <10% over the most global ocean (Fig. 4a). The Dust or BB source is the largest source of soluble iron deposition over the global ocean in the BASE simulation: the fraction of AN-, BB-, and Dust-dominant ocean area is 0.20%, 25%, and 75%, respectively (Fig. 4c). In contrast, in the NEW simulation, AN iron contributes to more than 20% of total soluble iron deposition flux over many parts of the ocean, except for the Atlantic, the Eastern Equatorial Pacific, and around Africa (Fig. 4b). AN iron is the largest source of soluble iron deposition, over 18% of the global ocean and 39% of the southern oceans (30–90 °S), in the NEW simulation (Fig. 4d). The fraction of the AN-dominant area (39%) is comparable to that of the Dust-dominant area (46%) over the southern oceans.

We examine the sensitivity of the simulated soluble iron deposition flux to the assumed magnetite fraction within the plausible range of 20–80%. The larger magnetite fraction lowers the contribution of AN iron to total soluble iron deposition flux.

Even if the magnetite fraction is assumed to be 80%, the contribution from AN source is greater than 20% over 28% of the global ocean and 53% of the southern oceans (30–90°S) (Supplementary Fig. 12c). In contrast, if we assume that the magnetite fraction is 20%, AN iron is the largest source of soluble iron deposition flux over 39% of the global ocean and 75% of the southern oceans (30–90 °S) (Supplementary Fig. 12f). These results show that AN iron will be an important source of soluble iron deposition regardless of the assumption of the magnetite fraction (i.e., even when the magnetite fraction is 80%), while the contribution of AN iron to total iron is sensitive to the assumption of the magnetite fraction.

## Discussion

AN magnetite DRF is estimated to be 0.021 W m$^{-2}$ globally and 0.22 W m$^{-2}$ over East Asia. Though the remaining uncertainty is large, we estimate that AN magnetite DRF can be comparable to BrC DRF. The global emission flux and the atmospheric burden of AN iron should be 5.0 and 7.7 times (at least 2.5 and 3.8 times) greater than the previous estimates, respectively, to explain the observed magnetite concentrations[17] (Supplementary Fig. 2). Our new AN iron emissions improve the agreement of total iron concentrations between measurements and model simulations (Supplementary Figs. 4 and 5) and increase the total soluble iron deposition flux by 52% over southern oceans

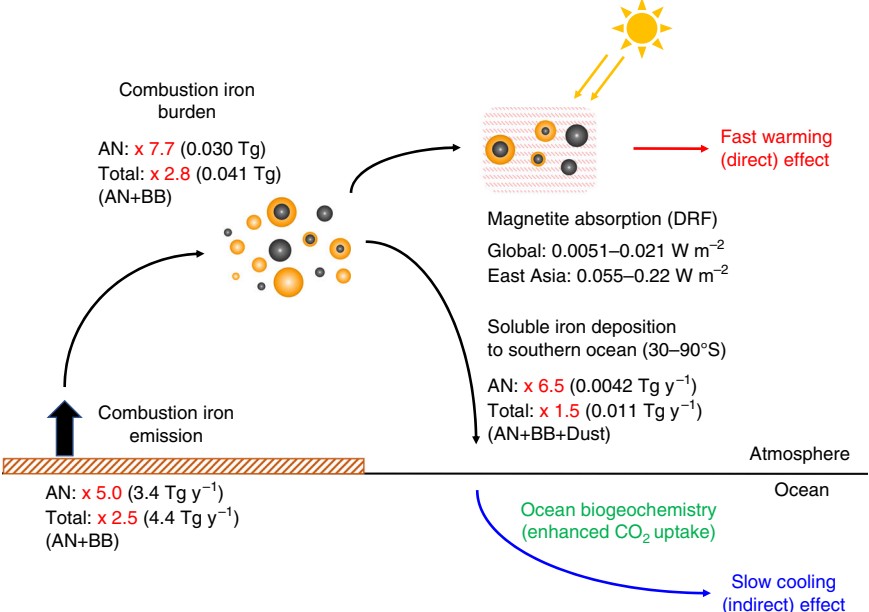

**Fig. 5** Schematic figure for the findings in this study. The values shown by black are global or regionally averaged statistics of combustion iron emission and burden, magnetite direct radiative forcing (DRF), and soluble iron deposition flux to southern oceans in the NEW simulation for anthropogenic (AN) and all sources (Total). The values shown by red are enhancement ratios in the NEW simulation from the BASE simulation

(30–90°S), where ocean primary productivity is iron limited, with the result that AN iron is the largest contributor (39%) to the total soluble iron deposition over these oceans (27% and 34% for mineral dust and biomass burning sources, respectively). Contrary to the previous expectations, our results strongly suggest that AN iron is an important player in the Earth's climate, through the heating of the atmosphere in a shorter timescale (hours to weeks[40]), as well as the cooling impact by modulating the ocean biogeochemistry in a much longer timescale (months to thousands of years[40]) (Fig. 5). The positive DRF would be much larger when we consider non-magnetite iron oxides (e.g., hematite). Long-term Earth system model simulations, considering our findings with the interactions between the atmosphere and the ocean including ocean bio-geochemistry, are necessary to understand the overall impact of AN iron on climate.

Since many factors contribute to the large uncertainties in the estimation of magnetite DRF and soluble iron deposition flux to the oceans, diverse observational data will be required to constrain them (Supplementary Note 1). First, simultaneous measurements of magnetite and total iron are needed to reduce the uncertainty in the fraction of magnetite to AN iron. Second, more field and laboratory experiments are needed to better understand the processes that control solubility of iron. Third, reducing the uncertainties in the imaginary part of the refractive index for magnetite and BrC is necessary to reduce the radiative forcing uncertainty. Fourth, due to the logistical constraints of ship-borne measurements, most available iron data have a limited time coverage over the oceans. Longer-term observations are needed to better estimate the iron amounts, especially within southern oceans. Fifth, due to the limitation of the instrument (Methods), magnetite particles greater than 3 μm are not con-sidered in our new estimates, and further measurements of coarse magnetite particles are necessary. Finally, more data on magnetite and combustion iron are needed, especially within Southern Africa, Australia, and their outflow regions. Data collection should ideally be focused on constraining uncertainties in soluble iron deposition flux to southern oceans with respect to its source contributions (AN or BB; Supplementary Note 2) in order to

quantify the impact of AN activity on biogeochemistry and $CO_2$ uptake over the southern oceans.

Since magnetite DRF has clear contrasts between land and ocean and between the Northern and Southern Hemispheres (Fig. 2a), magnetite may enhance the interactions of absorbing aerosols with atmospheric large-scale circulation systems and associated precipitation pattern, such as the Hadley circulation, the Intertropical Convergence Zone (ITCZ), and the Asian monsoon system[41–43]. It has been hypothesized that increased BC emissions contribute to the shifting of the ITCZ northward[41–43]. AN magnetite may further contribute to strengthening such absorbing aerosol-dynamical feedbacks.

Mahowald et al.[44] estimated that ocean primary productivity was enhanced by 6% due to doubling of desert dust during the 20th century. When we use this change in dust estimated by Mahowald et al.[44] along with our new estimates for AN iron, the PI to PD increase in the soluble iron deposition flux by AN iron exceeds that by iron in dust over 40% of the global ocean and 73% of the southern oceans (30–90 °S) (Fig. 6b). This suggests that AN iron is an important contributor to the PI to PD increase of soluble iron deposition, contrary to the conventional results (BASE simulation) that iron in dust is the dominant contributor (Fig. 6a). Our results suggest that AN iron will play a key role in various components in the Earth system (atmospheric aerosols, radiation, and dynamics, ocean biogeochemistry, global carbon cycle, and snow/ice albedo feedback) for both short and long timescales from the past to the future. Our findings will be important for making effective climate change policy in the future, as the reduction of anthropogenic combustion iron in the future will contribute to improve the air quality and cool the atmosphere from the regional-scale point of view (e.g., in mega-city regions), whereas it has a potential to increase the atmo-spheric $CO_2$ and hence heat the atmosphere by weakening ocean primary productivity and carbon uptake in a global perspective.

## Methods

**Global model simulations.** We conducted global model simulations using the Community Atmosphere Model version 5 (CAM5)[45–47] with the Aerosol Two-

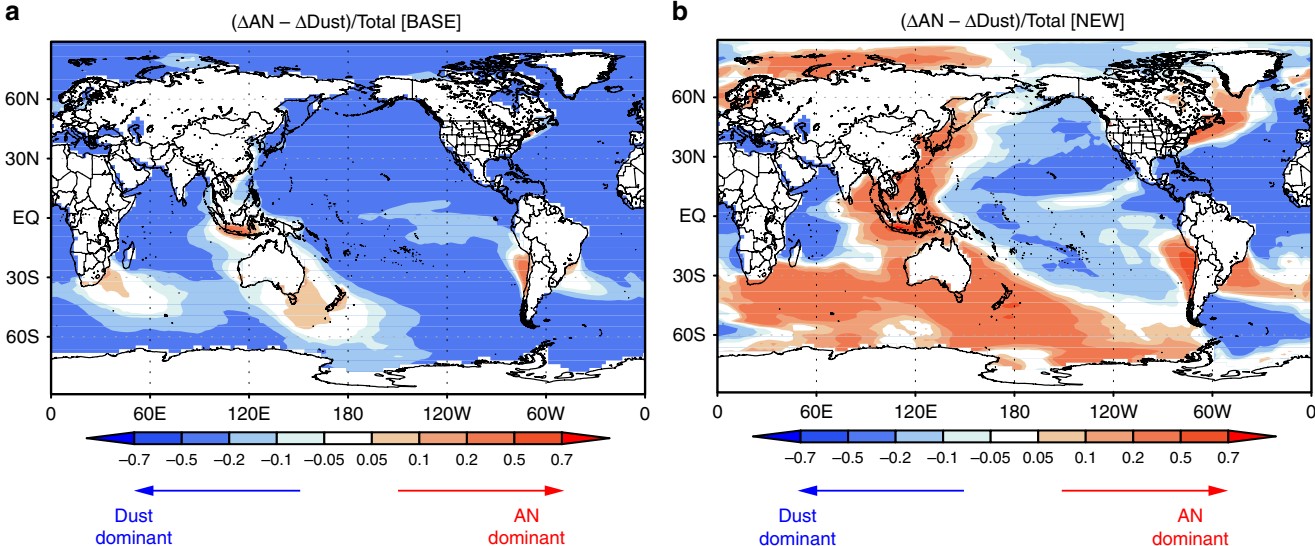

**Fig. 6** Relative importance of anthropogenic combustion iron and iron from mineral dust. **a, b** The preindustrial to present-day changes of soluble iron deposition flux were calculated for anthropogenic combustion iron (ΔAN) and iron from dust (ΔDust), and their differences were normalized by total deposition flux of soluble iron for present day (Total) for the **a** BASE and **b** NEW simulations. ΔDust was assumed to be half of the soluble iron deposition flux from Dust in present day[44]. Red regions show that anthropogenic combustion iron is more important than iron in dust, in terms of the preindustrial to present-day increases of the soluble iron deposition flux

dimensional bin module for foRmation and Aging Simulation version 2 (CAM5-chem/ATRAS2)[18,19] with modifications for magnetite. The standalone version of CAM5 (FC5 compset) in the Community Earth System Model (CESM) version 1.2.0 was used with horizontal resolution of 1.9° × 2.5° and 30 vertical layers from the surface to ~40 km. Online simulations were conducted for eight (2 × 2 × 2) cases: Two combustion iron emissions (BASE and NEW), 2 sources (AN only and AN+BB for combustion iron), and 2 periods (PD and PI). Each simulation was performed for six years, and averages of the last five years were used for analysis. Only emission data are different for all the eight cases. PI simulations were conducted with PD climatological data for sea surface temperature and sea ice[48]. We used anthropogenic, biomass burning, and biogenic emissions made by Lamarque et al.[49] for the years 2000 (PD) and 1850 (PI) for gas-phase and aerosol species except combustion iron (and magnetite). Dust emission flux was calculated online based on the scheme made by Zender et al.[22] with the modifications of Albani et al.[23] and size distributions of Kok[24]. Sea salt emissions were also calculated online[50,51]. Dust and sea salt emissions are almost the same between the PD and PI simulations because their changes from PI to PD (e.g., by changes in land use and climate) are not considered in this study. Simulations with magnetite from AN sources only were used to calculate the heating effect of anthropogenic magnetite. Simulations with AN+BB sources were used to estimate total iron concentrations and deposition flux of soluble iron.

Magnetite was added to the CAM5-chem/ATRAS2 model as a new aerosol species in this study. Nine aerosol species (sulfate, nitrate, ammonium, dust, sea salt, primary and secondary organic aerosol, BC, magnetite, and water) were treated in the model. The model considered emissions, gas-phase chemistry[52], condensation/evaporation of inorganic and organic species[18,53], coagulation[18,54,55], nucleation[56–58], activation of aerosol and evaporation from cloud[59,60], aerosol formation in clouds[61], dry and wet deposition[22,47,62,63], aerosol optical properties[18], aerosol-radiation interactions[64], and aerosol-cloud interactions[65,66]. Aerosol particles were resolved with 12 size bins from 0.001 to 10 μm in dry diameter. This representation corresponds to ATRAS-b in Matsui and Mahowald[19]. Particles in each size bin were assumed to be internally mixed. The values of density and refractive index were updated for BC and magnetite based on the values used in Moteki et al.[17].

In the BASE simulations, the flux and size distribution of combustion iron emissions (AN+BB) are based on the treatment of Luo et al.[1] and Mahowald et al.[67]. Coal combustion (power plant and industry) and residential heating are the main sources of combustion iron from AN (fossil fuel and biofuel) sources in these emission data. Since there is no information about iron speciation in these data, we cannot estimate the contribution of magnetite to total combustion iron emissions. We assumed all combustion iron had the density and refractive index of magnetite in the BASE simulations. This assumption does not affect the simulations because magnetite burden and DRF are small in the BASE simulations. The PI combustion iron emissions from AN sources were assumed to be 10% of PD emissions, following the assumption of Luo et al.[1]. We assumed no changes of combustion iron emissions between PI and PD for BB sources.

In the NEW simulations, the emission flux of AN magnetite was assumed to be double the AN iron emission flux used in Luo et al.[1] (BASE simulation). This factor

was chosen to make the model simulations consistent with the observed magnetite/BC mass ratio (0.40) over East Asia[17] (Supplementary Fig. 2a). The factor was applied globally by assuming the same spatial patterns for AN magnetite emissions (NEW) and AN iron emissions made by Luo et al.[1] (BASE). The size distribution of AN emissions was also modified based on the observed size distribution of magnetite (power function)[17]. This size distribution is smaller than that in the BASE simulations. The power function assumes no emissions of magnetite particles greater than 3 μm in diameter because the current single-particle soot photometer (SP2) cannot measure magnetite particles greater than 2.1 μm in diameter[17]. In addition, the power function underestimates the magnetite particle concentrations above ~0.6 μm because of the sampling loss[17]. Therefore, mass concentrations of magnetite are likely underestimated in the AN emissions for the NEW simulations. Only magnetite is treated as combustion iron from AN sources in the online simulations. Non-magnetite combustion iron from AN sources was estimated by offline calculations, with the assumption that 40% of combustion iron from AN sources is magnetite (see Supplementary Note 1). Regarding BB sources, the flux and size distribution of combustion iron emissions are the same between the BASE and NEW simulations. Therefore, only the treatment of combustion iron emissions from AN sources (flux and size distribution) is different between the BASE and NEW simulations. Similar to the BASE simulations, PI emissions for AN iron were assumed to be 10% of PD emissions in the NEW simulations.

The snow radiative forcing was estimated by Snow, Ice, and Aerosol Radiative (SNICAR) model[68] in the Community Land Model[69] (version 4.0). The snow radiative forcing was calculated for BC, BrC, and magnetite (for the NEW simulation in PD), as the difference in net shortwave radiative flux at the surface with all species present and with the target species (e.g. magnetite) excluded. The size distributions of BC and BrC in snow are the same as those treated by Flanner et al.[70]. The size distribution of magnetite in snow is assumed to be the power function, proposed by Moteki et al.[17], identical to the size distribution assumed for emissions. The complex refractive indices of BC and magnetite are assumed to be the values used in Moteki et al.[17]. The refractive index of BrC is assumed to be that of the moderately absorbing BrC, defined by Feng et al.[32]. Snow radiative forcing was calculated by assuming externally mixed spherical BC, BrC, and magnetite particles. We increased the forcing of BC and magnetite by a factor of 1.5 considering the absorption enhancement effect by coating species (described later).

**Heating effects of absorbing aerosols**. We used DRF for BC, BrC, and magnetite. DRF was calculated for all sky condition[71] as the difference of direct radiative effect (DRE) between the PD and PI simulations. For BC and magnetite, DRE was defined as the difference of shortwave radiative flux at the top of atmosphere between when all species were considered and when BC or magnetite was excluded. BrC DRE was defined as the difference of shortwave radiative flux at the top of atmosphere when all species were considered (with BrC absorption) and when BrC was assumed to be non-absorbing species (without BrC absorption). Though the definition is not the same between BC/magnetite and BrC, the definition of BrC DRE is consistent with previous studies focused on BrC[32,72].

The well-mixed mixing state assumption was used in the base optical calculations in the online simulations (ATRAS-b of Matsui and Mahowald[19]). In

this study, we did not use these calculations considering the following two points. First, absorption is severely overestimated when we use the well-mixed treatment for particles which contain a very small amount of absorbing species (e.g., in case of magnetite with the internally mixed treatment in ATRAS-b). Second, the base calculations do not consider non-spherical BC and magnetite particles. Considering these points, we conducted nine additional diagnostic optical and radiative transfer calculations in the online simulations to estimate DRF for BC, BrC, and magnetite. In these diagnostic calculations, we assumed that BC, BrC, magnetite, dust, and the others (inorganics, non-absorbing organics, and sea salt) were externally mixed (e.g., pure BC, pure magnetite). Number concentrations of the externally mixed particles were calculated from volume concentrations (mass/density) of each species and dry diameter in each size bin. Total number and mass concentrations (sum of all species) in the externally mixed treatment are identical to those in the base calculations. BrC, dust, and non-absorbing particles were assumed to be the mixture of each species and aerosol water which was calculated from the κ-Köhler theory[73]. The extinction coefficient, single scattering albedo, and asymmetry factor were calculated for 16 shortwave wavelengths from the look-up tables for the well mixed treatment[18], and they were introduced to the radiative transfer model[64]. BC and magnetite particles were assumed to have no water. The look-up tables considering the effects of non-sphericity of BC and magnetite to their light-absorbing and scattering efficiencies were prepared according to the methods described in Moteki et al.[17]. Non-sphericity of magnetite increases the global-mean absorption aerosol optical depth (at 0.55 μm) by 65% (compared with the spherical treatment). To consider the enhancement of absorption by coating species, we enhanced DRF (for externally mixed treatment) by 50% for BC and magnetite. This factor was obtained in our previous global model simulations which resolved BC mixing state in detail[19].

BrC emissions and formation processes were not considered explicitly in this study. BrC was assumed to be 10% of total organic aerosols globally. The burden (0.29 Tg) and spatial distribution of BrC using this simple assumption is generally consistent with previous studies[32,72]. The imaginary part of BrC refractive index was obtained from the moderately and strongly absorbing BrC defined by Feng et al.[32]. BrC DRE in our simulations (in PD) is 0.066 and 0.027 W m$^{-2}$ for the strongly and moderately absorbing BrC, respectively. These values are a slightly smaller than the previous estimates (0.04–0.11 W m$^{-2}$)[32,72]. BrC DRF (0.016–0.038 W m$^{-2}$ in our simulations) is smaller than BrC DRE because DRF is the difference of DRE between PD and PI. BrC DRF estimated in this study have large uncertainties, but these BrC results are useful to discuss the relative importance of BrC and magnetite.

We used the observed magnetite/BC ratio to estimate magnetite emissions in the NEW simulations. The estimation of magnetite burden and DRF therefore depends on the accuracy of BC burden and DRF in the model. BC DRF in this study is 0.13 W m$^{-2}$. This value is smaller than the averages of other global models (e.g., 0.18 W m$^{-2}$ estimated in Myhre et al.[30]), but within the range of the global models (0.05–0.37 W m$^{-2}$)[31]. However, this DRF value is smaller than BC DRF estimates constrained by surface solar radiation observations[31,74] by more than a factor of 4 (e.g., 0.71 W m$^{-2}$ in the estimation of Bond et al.[31]). Considering this underestimation of BC DRF, we calculated four times greater magnetite DRF than the online simulation as the maximum estimate of magnetite DRF.

**Soluble iron estimation**. We estimated the deposition flux of soluble iron to the ocean from the BASE and NEW simulations. The deposition flux of soluble iron was estimated from the following equation:

$$\mathrm{DF}_{\mathrm{soliron},s,i,t} = \mathrm{DF}_{\mathrm{iron},s,i,t} \times \mathrm{f}_{\mathrm{sol},s,i,t} \quad (1)$$

where $\mathrm{DF}_{\mathrm{soliron},s,i,t}$ and $\mathrm{DF}_{\mathrm{iron},s,i,t}$ are the deposition flux of soluble iron and total iron, respectively, from source $s$ (AN, BB, or Dust) at horizontal grid $i$ and at time (month) $t$, and $f_{\mathrm{sol},s,i,t}$ is the soluble fraction of iron (soluble/total) for source $s$ at grid $i$ and time $t$. $\mathrm{DF}_{\mathrm{iron},s,i,t}$ was simulated in our model with the assumption that 3.5% of dust is iron[1,75]. Since the CAM5-chem/ATRAS2 model does not consider soluble iron formation processes, $f_{\mathrm{sol},s,i,t}$ was given from another global aerosol model which considers direct emissions and atmospheric formation processes (chemical and physical processes that are dependent on temperature, acidity of particles, and concentrations of organic acids) of soluble iron explicitly[36] (Supplementary Figs 11a–c). Magnetite and other iron species were assumed to have the same solubility. The global-mean iron solubility for each source (7.7%, 11%, and 4.0% for AN, BB, and Dust sources, respectively, over the ocean) is within the ranges of previous studies[10–12,36,76–78]. To add soluble iron processes and the diversity of iron-containing particles (e.g., particle size, mixing state, mineralogy)[79–82] to the CAM5-chem/ATRAS2 model may be an important future study.

**Comparison with magnetite measurements**. We used observed magnetite/BC mass ratios during the Aerosol Radiative Forcing in East Asia (A-FORCE) 2013W aircraft campaign[17,83] and intensive surface measurements at Zeppelin (Supplementary Fig. 2). BC and magnetite particles were measured by an SP2 with the detectable size ranges of 0.070–0.85 μm and 0.17–2.1 μm (mass-equivalent diameters), respectively[17]. Since the observation system may underestimate magnetite particles greater than 0.6 μm (up to by a factor of 2 at 2.1 μm)[17], magnetite concentrations in the atmosphere may be higher than the observed concentrations.

The A-FORCE campaign was conducted over the Yellow Sea and the East China Sea in February and March (14 February to 10 March) 2013[17,83] to observe the outflows from anthropogenic combustion sources in East Asia. Measurements at Zeppelin (78.9°N, 11.9°E, 474 m above sea level) were conducted in March (28 February to 29 March) 2017. The observation site is representative of European Arctic background and can be influenced by pollutants transported from Europe and Russia[84,85]. The observed number and mass concentrations of magnetite correlated with those of BC, with the square of the correlation coefficient ($R^2$) of 0.90 and 0.50, respectively (Supplementary Fig. 13). These results suggest that emission sources of the observed magnetite and BC were spatially similar, as is also seen in the Asian continental outflow[17] and urban sites in Japan (Ohata et al., in prep). Simulated magnetite/BC ratios calculated from 5-year averages for the observed months (February and March for A-FORCE and March at Zeppelin) were compared with observed magnetite/BC ratios. We used the region of 125°–137°E and 27°–33°N for comparisons with the A-FORCE campaign. The size ranges of BC and magnetite were chosen to be consistent with the observed size ranges. The spatial and/or temporal averages of the magnetite/BC mass ratio were calculated from the average of magnetite/BC ratios for both measurements and model simulations. Regions and periods are not exactly the same between measurements and model simulations. In addition, model simulations are not for specific years when measurements were conducted. However, these comparisons are useful to show the dramatically improved model performance in the NEW simulation (Supplementary Fig. 2).

**Code availability**. The codes used to conduct the analysis presented in this paper can be obtained by contacting the corresponding author (H.M.).

**Data availability**. Data used in Figs. 1–4 and Fig. 6 are available at http://has.env.nagoya-u.ac.jp/~matsui/data. Other data are available upon request from the corresponding author (H.M.)

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

## Acknowledgements

H.M. was supported by the Japan Society for the Promotion of Science (JSPS) Overseas Research Fellowships. This work was supported by the Ministry of Education, Culture, Sports, Science, and Technology and the Japan Society for the Promotion of Science (MEXT/JSPS) KAKENHI Grant Numbers JP26740014, JP17H04709, JP26241003, JP16H01770, and JP15H05465 and MEXT Green Network of Excellence (GRENE) and Arctic Challenge for Sustainability (ArCS) projects. This work was also supported by the global environment research funds of the Ministry of the Environment, Japan (2-1403, 2-1703). N.M.M., R.A.S., and D.S.H. acknowledge support from DOE-SC0006791 and SC0006735, NSF-1049033, high-performance computing support from Yellowstone (ark:/85065/d7wd3xhc) provided by NCAR's Computational and Information Systems Laboratory, sponsored by the National Science Foundation and Atkinson Center for a Sustainable Future.

## Author contributions

H.M. designed the research, performed model simulations and data analysis, and wrote the manuscript. H.M., N.M.M., and N.M. suggested analysis, interpreted the data, and discussed their implications. N.M., S.O., M.K., and A.Y. conducted magnetite and BC measurements during the A-FORCE 2013W campaign and at Zeppelin and suggested global evaluations of magnetite heating effect. N.M. and H.M. made look-up tables for optical properties of magnetite and BC. M.F. and H.M. made optical data of magnetite, BC, and BrC for the SNICAR. D.S.H., R.A.S., and H.M. made solubility data for soluble iron analysis. All authors commented on and contributed to the manuscript.

## Additional information

**Competing interests:** The authors declare no competing interests.

