## [Peer Review File · Nature Communications]

Reviewers' comments:

Reviewer #1 (Remarks to the Author):

Review of "Anthropogenic combustion iron as a complex climate forcer" by Metsui et al.

This manuscript presents modeling results of anthropogenic iron in the global atmosphere, and its potential contribution to direct radiative forcing and soluble iron flux to the ocean. The paper is well written and is based on an extensive amount of work. I would feel bad to object to its publication.

But I do like to point out a problem with the work. Previous studies of anthropogenic iron are based on extensive measurements of total iron concentrations in aerosol or iron-to-black carbon ratio (Fe/BC) in emissions, such as Luo et al. (2008). The new study by Matsui et al. assumes these measurements (Fe/BC) are "incorrect", and should be corrected/scaled by relatively few measurements of (magnetite Fe)/BC ratio and uncertain (total Fe)/(magnetite Fe) ratio. This feels like building a house on sand. The study would be more valid had it shown these new measurements could be reasonably extrapolated from East Asia to Africa and Australia.

On the other hand, the paper, if published, could inspire targeted global measurements of aerosol iron, especially over the Southern Oceans where measurements are scarce.

Reviewer #2 (Remarks to the Author):

This a very interesting manuscript giving first estimates of the magnitude of radiative forcing of anthropogenic combustion iron. It is appropriate for Nature Communications and should be published after the following comment is taken into account:

I believe that the error range of the forcing results is greatly understated and the utility of this manuscript is less in estimating actual combustion iron forcings but more in pointing out what information is needed for arriving at such forcings. While the Supplementary Information includes an interesting discussion of uncertainties in "1. Summary of uncertainties in this study", this yields less of an accurate estimation of uncertainties but more a list of data needs before the radiative forcing and its uncertainties can be estimated. Examples from the supplement include:

1. Estimation of the magnetite fraction is based on very few measurements for selected sources and it is unclear if the factor of four for the uncertainty really covers its range.
2. The spatial distribution of magnetite emissions is essentially unknown as there are "no magnetite data over North America and the southern hemisphere".
3. Again, very little is known about iron solubility and emissions and formation of soluble iron. The fact that also "other studies on soluble iron deposition to the ocean" are affected by this is not very comforting either.
4. Optical properties of magnetite and brown carbon. There has been very little work on the refractive index of magnetite and it is not clear how representative the two references that are given here are. In addition, the refractive index of brown carbon is based on a single 2013 reference and there are many more estimations out there (especially after 2013) that would show much greater uncertainty than represented here.
5. Comparisons with measurements are also problematic due to the limited time coverage over oceans.
6. Coarse magnetite particles represent an additional problem, due to the size limitations of the laser induced incandescence instrument used (DMT SP2).

While, the manuscript mentions most of these problems, I believe it greatly underestimates the uncertainty of the resulting modeling results because many of the individual uncertainties cannot

be easily quantified. Therefore, I recommend refocusing the manuscript on the data that are needed to quantify uncertainties and to do meaningful modeling.

Reviewer #3 (Remarks to the Author):

Summary:

Atmospheric deposition of aerosol iron is recognized to be a significant source of this micronutrient to large areas of the open ocean. This process potentially relieves iron limitation of primary productivity and therefore is linked to the global carbon cycle. The community is learning more about that this important process through both observational field studies and modeling studies and the two are closely linked. One key uncertainty, among many, is the relative proportion of more soluble anthropogenic aerosol iron to less soluble crustal aerosol iron. This manuscript takes recently published findings from East Asia and incorporates them into a global aerosol model. Their results suggest that current deposition estimates greatly underestimate the deposition of soluble iron to oceans in the Southern Hemisphere. An additional finding from this study is that anthropogenic magnetite aerosols exert a significant impact on direct radiative forcing. The conclusion is that anthropogenic aerosols have a complex impact on global climate. The results have large uncertainties which are discussed in the supplementary materials.

This manuscript requires significant revision to bolster its conclusions before it is accepted.

General Comments:

The authors have put together an interesting paper that attempts to address an important uncertainty. Collections of atmospheric aerosols remain relatively sparse and are not well distributed spatially underscoring the importance of using global scale models. Their use is particularly important for regions which lack long-term land island-based sampling like much of the Southern Hemisphere. Modeling studies should help field scientists direct their work and this study demonstrates that more attention is needed on the characterization of anthropogenic magnetite.

My primary concern about this study and the interpretations of the results lies in the use of relatively small datasets from East Asia and Northern Europe to constrain the amount of anthropogenic iron for the whole globe. Are samples from Asian outflow representative of aerosols for other regions in magnitude and composition? The issue is most stark for the Southern Ocean. Is it reasonable to think that the relative proportion of anthropogenic iron in Southern Hemisphere aerosols is similar to that in Asian aerosols? The authors use the observed magnetite to black carbon mass ratio for East Asia to estimate the emission flux under the NEW simulation conditions but there few observations of either in the Southern Hemisphere, a point they mention in the supplemental information. Where is all the modeled Southern Hemisphere anthropogenic iron coming from? The manuscript needs more discussion of potential emission sources in the Southern Hemisphere to support the modeling results.

On page 20, "intensive surface measurements at Zeppelin" are mentioned. These should be introduced more clearly, and that location placed in better context. The only information that is included is the location (79N) and duration of sampling (1 month). The supplementary information does not elaborate on the source of this data. How do these measurements fit in with the conclusions of the study? Are European emissions any more relevant to the Southern Hemisphere than Asian aerosols?

It may be that the authors have high confidence in the model results, for the Southern Hemisphere in particular, but the reasoning supporting that confidence is readily clear to me. I am not a modeler and could be missing an important aspect of that portion of the work, but then other readers may reach the same interpretations as I have. If this manuscript is revised, I encourage the authors to make the connections between the East Asian source data and the global outputs clearer.

Response to reviewer #1

NCOMMS-17-28537: “Anthropogenic combustion iron as a complex climate forcer” by H. Matsui et al.

We thank the reviewer very much for reading the paper carefully and giving us valuable comments. We revised the paper by taking into account the reviewer’s comments. In this revision, we clarified the validity of the new iron emissions (used in the NEW simulation) by comparing simulation results with total iron observations at Gosan (used in Luo et al. (2008)). We also added some results to support the validity of extending the magnetite/BC data in East Asia to Southern Africa and Australia.

Detailed responses to individual comments and suggestions are given below.

Reviewer’s comment:

Previous studies of anthropogenic iron are based on extensive measurements of total iron concentrations in aerosol or iron-to-black carbon ratio (Fe/BC) in emissions, such as Luo et al. (2008). The new study by Matsui et al. assumes these measurements (Fe/BC) are "incorrect", and should be corrected/scaled by relatively few measurements of (magnetite Fe)/BC ratio and uncertain (total Fe)/(magnetite Fe) ratio. This feels like building a house on sand.

Response:

We thank the reviewer for pointing out this important point. We compared our model simulations with total iron observations at Gosan in East Asia (used in Luo et al. (2008)). Supplementary Fig. 4 shows the consistency between Luo et al. (2008) and this study and the validity of the new AN iron emission estimates used in the NEW simulation which are estimated from the observed magnetite/BC ratio using an assumed magnetite/iron fraction of 40%. We added these results to the main text (Lines 123-131). Basically in Luo et al. (2008), we used concentration ratios to find emission ratios, which is not quite true, if the BC and iron have different sinks. Here we show that the new emission ratios better match the concentration ratios.

====

Lines 123-131: “Comparisons of model simulations with total iron (AN+BB+Dust) observations at Gosan in East Asia¹ show that the iron emissions used in the NEW simulation are more consistent with the measurements than those used in the BASE simulation and Luo et al.¹. The NEW simulation (annual mean total iron concentrations are $0.66 \mu\text{g m}^{-3}$) has a good agreement with the total iron observations¹ ($0.63 \mu\text{g m}^{-3}$), whereas simulations in the BASE case ($0.36 \mu\text{g m}^{-3}$) and Luo et al.¹ ($0.21 \mu\text{g m}^{-3}$) underestimate iron concentrations by a factor of 2 or 3 (Supplementary Fig. S4). Supplementary Fig. S4 also shows that the NEW case can simulate the observed iron concentrations in both summer and winter (when the contribution from Dust iron is limited, i.e. AN iron is dominant).”

Supplementary Figure S4. Monthly variations of total iron concentrations at Gosan, East Asia. Observed total iron data (black) from Fig. 6 of Luo *et al.* (2008). Simulated total iron results are shown for the BASE (blue) and NEW (red) simulations (5-year mean). Iron concentrations from dust sources only is shown for the BASE simulation (orange, 5-year mean).

Reviewer's comment:

The study would be more valid had it shown these new measurements could be reasonably extrapolated from East Asia to Africa and Australia.

Response:

We have strengthened the validity of extrapolating the magnetite results in East Asia to Southern Africa and Australia through additional two analyses using emission inventories and total iron observations. Results are summarized in the main text (Lines 107-113 and Lines 150-155). Details of the two analyses were added to Supplementary Discussion 2.

Lines 107-113: "Observations of magnetite concentrations (or magnetite/BC) are missing over North America and the Southern Hemisphere. In addition, emission sources of magnetite are currently not known well. However, the estimation of possible emission sources of magnetite from BC emission inventories and the underestimation of combustion iron around Southern Africa and Australia support the validity of extending the magnetite results in East Asia to the Southern Hemisphere where we focus on in soluble iron estimates (Supplementary Discussion 2)."

Lines 150-155: "Since combustion iron is dominant and iron/BC ratios are underestimated around Southern Africa and Australia in our simulations, combustion iron emissions (from AN or BB sources, or a combination of both) are likely underestimated over these regions (Supplementary Discussion 2.2). This suggests that the underestimation of AN iron emissions is a main source of the underestimation of total iron around Southern Africa and Australia (Supplementary Discussion 2.2)."

Supplementary Discussion 2:

2. Extension of magnetite results in East Asia to the Southern Hemisphere

Since observations of magnetite concentrations (or magnetite/BC) are missing in the Southern Hemisphere, it is not currently possible to show direct evidence as to whether we can extend the magnetite results in East Asia to the Southern Hemisphere (Southern Africa and Australia). However, we show the validity of the extension to the Southern Hemisphere indirectly from the following two analyses using emission inventories and total iron observations.

2.1. Potential emission sources of magnetite

First, we examine potential emission sources of magnetite over East Asia, Europe, Southern Africa, and Australia by using the BC emission inventory used in this study (*Lamarque et al.*, 2010). The 4 results below (a-d) suggest that the extrapolation of the magnetite results in East Asia and Europe (Supplementary Fig. S2) to Southern Africa and Australia is a more reasonable assumption than applying the magnetite results to the Northern Hemisphere only.

- a. Magnetite and BC observations show that these two species have similar temporal variations in both East Asia (*Moteki et al.*, 2017) and Europe (Supplementary Fig. S13 at Zeppelin). Though emission sources of magnetite are not known well, this result suggests that magnetite and BC have similar emission sources (from fossil fuel combustion).
- b. Observed magnetite/BC ratios in East Asia and Europe are reproduced better in the NEW simulation (Supplementary Fig. S2). This result suggests that magnetite emissions used in the NEW simulation are reasonable over East Asia and Europe.
- c. BC from fossil fuel combustion is emitted mainly from the industry and transport sectors (*Lamarque et al.*, 2010). In East Asia (100°-140°E and 20°-45°N), industry sector is dominant (industry and transport sectors account for 78% and 18% of total BC emissions from fossil fuel combustion, respectively). In Europe (0°-60°E and 35°-70°N), transport sector is dominant (industry and transport sectors account for 31% and 62% of total BC emissions from fossil fuel combustion, respectively). These results suggest that industry and transport sectors may be main sources of magnetite and that the new magnetite emissions used in the NEW simulation are suitable for both the industry and transport sectors.
- d. BC emissions from industry and transport sectors are comparable over Southern Africa (0°-60°E and 0°-40°S, 44% from industry and 42% from transport) and Australia (110°-180°E and 10°-50°S, 37% from industry and 46% from transport). The two sectors account for 87% and 84% of total BC emissions from fossil fuel combustion over Southern Africa and Australia, respectively. These results suggest the extension of the results in East Asia (industry sector dominant) and Europe (transport sector dominant) to Southern Africa and Australia is a reasonable assumption.

2.2. Underestimation of combustion iron over Southern Africa and Australia

Second, we compare model simulations with BC and total iron observations in the Southern Hemisphere. The 5 results below (e-i) suggest that combustion iron (AN or BB sources or both) over Southern Africa and Australia and their marine outflow regions is underestimated in the model simulations and that the agreement with the iron observations is improved by extending the magnetite results in East Asia to the Southern

Hemisphere (NEW simulation).

- e. BC concentrations near the surface are well simulated (or slightly overestimated) in the Southern Hemisphere in our model, as shown by Figs 9g-9h of *Matsui and Mahowald (2017)*. This result suggests that BC emissions and concentrations are valid in the southern hemisphere.
- f. In contrast, surface iron concentrations and its deposition flux are underestimated around Southern Africa and Australia (Supplementary Figs S6-S7). The iron/BC emissions ratio is therefore likely underestimated within these Southern Hemisphere regions.
- g. The contribution of mineral dust to total iron concentrations is low (<50%) around Southern Africa and Australia (Supplementary Figs S7c-S7d). This result suggests that combustion iron emissions (AN and/or BB) are underestimated within these regions.
- h. Simulated iron concentrations are comparable to observations around Amazon where the BB contribution is high (Supplementary Fig. S7e), suggesting the following two possibilities. One is the underestimation of the iron/BC ratio in AN emissions within Southern Africa and Australia. The other is the underestimation of the iron/BC ratio in BB emissions within Southern Africa and Australia because the iron/BC ratio within these regions may not be the same as (higher than) that over Amazon (the iron/BC ratio in BB emissions is assumed to be the same over Amazon and over Southern Africa and Australia in our simulations). One of these two possibilities or both contribute to the underestimation of combustion iron around Southern Africa and Australia.
- i. The NEW simulation has better agreement with the observations of iron concentrations and deposition flux than the BASE simulation around Southern Africa and Australia (Supplementary Fig. S7e). This suggests AN iron emissions in the NEW simulation are more realistic than those in the BASE simulation.

Further magnetite and combustion iron observational data are needed in the Southern Hemisphere, particularly downwind of Southern Africa and Australia. Furthermore, attribution of combustion iron to AN and BB sources is needed to quantify the relative contribution of these sources. However, the results shown above suggest that the underestimation of AN iron emissions is a main source of the underestimation of combustion iron concentrations around Southern Africa and Australia.

Supplementary Figure S6. Deposition flux of total iron. Observed and simulated total iron deposition flux for the BASE (blue) and NEW (red) simulations. Observed data are derived from *Mahowald et al. (2009)*.

Supplementary Figure S7e. The ratio between simulated and observed total iron concentrations in the Southern Hemisphere (5-year mean).

Supplementary Figure S13. Scatterplots between magnetite and BC concentrations at Zeppelin. a-b, Scatterplots of hourly observed data for (a) number and (b) mass concentrations.

===

Reviewer's comment:

On the other hand, the paper, if published, could inspire targeted global measurements of aerosol iron, especially over the Southern Oceans where measurements are scarce.

Response:

We appreciate the reviewer's positive suggestion. We agree this discussion is an important guideline for researchers related to iron observations and a main conclusion and interpretation of this study. We added the following description to the main text (Lines 290-295).

====

Lines 290-295: "More data on magnetite and combustion iron are needed, especially within Southern Africa and Australia and their outflow regions. Data collection should ideally be focused on constraining uncertainties in soluble iron deposition flux to southern oceans with respect to its source contributions (AN or BB; Supplementary Discussion 2.2) in order to quantify the impact of AN activity on biogeochemistry and CO₂ uptake over the southern oceans."

====

Response to reviewer #2

NCOMMS-17-28537: “Anthropogenic combustion iron as a complex climate forcer” by H. Matsui et al.

We thank the reviewer very much for reading the paper carefully and giving us valuable comments. We revised the paper by taking into account the reviewer’s comments. In this revision, we added discussions on what observations are needed for reducing the uncertainties in magnetite DRF and soluble iron deposition flux.

Detailed responses to individual comments and suggestions are given below.

Reviewer’s comment:

I believe that the error range of the forcing results is greatly understated and the utility of this manuscript is less in estimating actual combustion iron forcings but more in pointing out what information is needed for arriving at such forcings. While the Supplementary Information includes an interesting discussion of uncertainties in “1. Summary of uncertainties in this study”, this yields less of an accurate estimation of uncertainties but more a list of data needs before the radiative forcing and its uncertainties can be estimated.

Response:

Considering this important comment by the reviewer, we added a paragraph to the main text (Lines 277-295) to highlight important observations needed in the future to help reduce the uncertainties in the estimation of both the magnetite DRF and soluble iron deposition flux to oceans. Responses to the 6 points the reviewer raised are given below individually.

====
Lines 277-295: “Since many factors contribute to the large uncertainties in the estimation of magnetite DRF and soluble iron deposition flux to the oceans, diverse observational data will therefore be required in order to constrain them (Supplementary Discussion 1). First, simultaneous measurements of magnetite and total iron are needed to reduce the uncertainty in the fraction of magnetite to AN iron. Second, more field and laboratory experiments are needed to better understand processes which control solubility of iron. Third, reducing the uncertainties in the imaginary part of the refractive index for magnetite and BrC is necessary to reduce radiative forcing uncertainty. Fourth, due to the logistical constraints of ship-borne measurements, most available iron data have a limited time coverage over oceans. Longer-term observations are needed to better estimate iron amounts, especially within southern oceans. Fifth, due to the limitation of the instrument (Methods), magnetite particles greater than 3 μm are not considered in our new estimates and further measurements of coarse magnetite particles are necessary. Finally, more data on magnetite and combustion iron are needed, especially within Southern Africa and Australia and their outflow regions. Data collection should ideally be focused on constraining uncertainties in soluble iron deposition flux to southern oceans with respect to its source contributions (AN or BB; Supplementary Discussion 2.2) in order to quantify the impact of AN activity on biogeochemistry and CO_2 uptake over the southern oceans.”

====

Reviewer's comment:

1. Estimation of the magnetite fraction is based on very few measurements for selected sources and it is unclear if the factor of four for the uncertainty really covers its range.

Response:

As the reviewer pointed out, it is not clear whether the factor 4 uncertainty for the magnetite fraction covers the full uncertainty range. We added results showing the validity of our new AN iron emissions (used in the NEW simulation) to the main text (Lines 123-131) by comparing model simulations with total iron measurements at Gosan in East Asia.

We also added the following description to the main text (Lines 280-281) and Supplementary Discussion 1a: "Simultaneous measurements of magnetite and total iron are needed to reduce the uncertainty in the fraction of magnetite to AN iron and its spatial and temporal variations."

====

Lines 123-131: "Comparisons of model simulations with total iron (AN+BB+Dust) observations at Gosan in East Asia¹ show that the iron emissions used in the NEW simulation are more consistent with the measurements than those used in the BASE simulation and Luo et al.¹. The NEW simulation (annual mean total iron concentrations are $0.66 \mu\text{g m}^{-3}$) has a good agreement with the total iron observations¹ ($0.63 \mu\text{g m}^{-3}$), whereas simulations in the BASE case ($0.36 \mu\text{g m}^{-3}$) and Luo et al.¹ ($0.21 \mu\text{g m}^{-3}$) underestimate iron concentrations by a factor of 2 or 3 (Supplementary Fig. S4). Supplementary Fig. S4 also shows that the NEW case can simulate the observed iron concentrations in both summer and winter (when the contribution from Dust iron is limited, i.e. AN iron is dominant)."

Supplementary Figure S4. Monthly variations of total iron concentrations at Gosan, East Asia. Observed total iron data (black) from Fig. 6 of Luo et al. (2008). Simulated total iron results are shown for the BASE (blue) and NEW (red) simulations (5-year mean). Iron concentrations from dust sources only is shown for the BASE simulation (orange, 5-year mean).

====

Reviewer's comment:

2. The spatial distribution of magnetite emissions is essentially unknown as there are “no magnetite data over North America and the southern hemisphere”.

Response:

We agree that more magnetite data are needed, especially over the Southern Hemisphere, as the reviewer pointed out. We added the following description to Supplementary Discussion 1b to clarify this point: “We showed simulated total iron concentrations were generally consistent with measurements in Supplementary Figs S4-S7. We also showed the validity of extending the magnetite results in East Asia to the Southern Hemisphere indirectly through the estimation of possible emission sources of magnetite and the underestimation of combustion iron around Southern Africa and Australia (Supplementary Discussion 2). However, more data on magnetite and combustion iron are needed, especially within Southern Africa and Australia and their outflow regions. Data collection should ideally be focused on constraining uncertainties in soluble iron deposition flux to southern oceans with respect to its source contributions (AN or BB; Supplementary Discussion 2.2) in order to quantify the impact of AN activity on biogeochemistry and CO₂ uptake over the southern oceans.” This description is an important guideline for researchers related to iron observations and a main conclusion and interpretation of this study.

We also added summary of this description to the main text (Lines 290-295).

Reviewer's comment:

3. Again, very little is known about iron solubility and emissions and formation of soluble iron. The fact that also “other studies on soluble iron deposition to the ocean” are affected by this is not very comforting either.

Response:

We added the following description to Supplementary Discussion 1c: “More field and laboratory experiments are needed to better understand processes which control solubility of iron and its spatial and temporal variability in the atmosphere. Numerical models which can represent these processes also have to be developed and evaluated.”

We also added the summary of this description to the main text (Lines 281-282).

Reviewer's comment:

4. Optical properties of magnetite and brown carbon. There has been very little work on the refractive index of magnetite and it is not clear how representative the two references that are given here are. In addition, the refractive index of brown carbon is based on a single 2013 reference and there are many more estimations out there (especially after 2013) that would show much greater uncertainty than represented here.

Response:

Considering the reviewer's comment, we added the following description to the Supplementary Discussion 1d: "As a wavelength-dependent complex refractive index of magnetite, we have adopted the experimental data for pure magnetite provided by *Huffman and Stapp* (1973), which was referenced in Supplementary Information of *Moteki et al.* (2017). To our knowledge, this is only the publicly-available source of wavelength-dependent optical constant of magnetite. At this point, however, we are not able to assess how this data represent the optical constants of combustion-induced aerosol magnetite."

We also added the following description for the uncertainty in the refractive index of BrC to Supplementary Discussion 1d. "The range of the refractive index (imaginary part) of BrC in *Feng et al.* (2013) is 0.075-0.168 at 350 nm and 0.02-0.063 at 450 nm. These values were estimated to match the observed absorption cross sections in *Chen and Bond* (2010) and *Kirchstetter et al.* (2004). Some studies, however, reported much lower imaginary part of the refractive index of BrC: 0.0082 at 350 nm (*Lack et al.*, 2012), 0.009 at 450 nm (*Liu et al.*, 2013), and 0.042 at 365 nm (*Shamjad et al.*, 2016). This suggests that the imaginary part of the refractive index of BrC may be more uncertain than the range of strongly and moderately absorbing BrC in *Feng et al.* (2013). BrC may have a lower DRF when these refractive indices are used in our simulations. Reducing the uncertainties in the imaginary part of the refractive index for magnetite and BrC is therefore important."

The summary of this description was added to the main text (Lines 174-177) as follows: "Since some recent studies reported lower imaginary part of the refractive index of BrC than the moderately absorbing BrC (Supplementary Discussion 1d), BrC could have a lower DRF when these refractive indices are used in our simulations."

Reviewer's comment:

5. Comparisons with measurements are also problematic due to the limited time coverage over oceans.

Response:

We added the following description to the main text (Lines 284-287) and Supplementary Discussion 1e: "Due to the logistical constraints of ship-borne measurements, most available iron data have a limited time coverage over oceans. Longer-term observations are needed to better estimate iron amounts, especially within southern oceans."

Reviewer's comment:

6. Coarse magnetite particles represent an additional problem, due to the size limitations of the laser induced incandescence instrument used (DMT SP2).

Response:

We added the following description to the main text (Lines 287-289) to clarify the importance of measurements for coarse magnetite particles: "Due to the limitation of the instrument (Methods), magnetite particles greater than 3 μm are not considered in our new estimates and further measurements of coarse magnetite particles are necessary."

Reviewer's comment:

While, the manuscript mentions most of these problems, I believe it greatly underestimates the uncertainty of the resulting modeling results because many of the individual uncertainties cannot be easily quantified. Therefore, I recommend refocusing the manuscript on the data that are needed to quantify uncertainties and to do meaningful modeling.

Response:

As described in the response to the first comment, we added a paragraph to the Discussion section to highlight important observations for future studies to reduce the uncertainties in the estimation of magnetite DRF and soluble iron deposition flux to oceans and its source contributions.

Response to reviewer #3

NCOMMS-17-28537: “Anthropogenic combustion iron as a complex climate forcer” by H. Matsui et al.

We thank the reviewer very much for reading the paper carefully and giving us valuable comments. We revised the paper by taking into account the reviewer’s comments. In this revision, we added some results and interpretations to support the validity of extrapolating the magnetite observations in East Asia to Southern Africa and Australia where magnetite observations are not available currently.

Detailed responses to individual comments and suggestions are given below.

Reviewer’s comment:

My primary concern about this study and the interpretations of the results lies in the use of relatively small datasets from East Asia and Northern Europe to constrain the amount of anthropogenic iron for the whole globe. Are samples from Asian outflow representative of aerosols for other regions in magnitude and composition? The issue is most stark for the Southern Ocean. Is it reasonable to think that the relative proportion of anthropogenic iron in Southern Hemisphere aerosols is similar to that in Asian aerosols? The authors use the observed magnetite to black carbon mass ratio for East Asia to estimate the emission flux under the NEW simulation conditions but there few observations of either in the Southern Hemisphere, a point they mention in the supplemental information. Where is all the modeled Southern Hemisphere anthropogenic iron coming from? The manuscript needs more discussion of potential emission sources in the Southern Hemisphere to support the modeling results.

Response:

We thank the reviewer for pointing out this important point. Since there are no magnetite (or magnetite/BC) observations available in the Southern Hemisphere, it is not currently possible to show direct evidence as to whether we can extend the magnetite results in East Asia to the Southern Hemisphere. However, two analyses using emission inventories and total iron observations do support the validity of extending the magnetite results in East Asia to the Southern Hemisphere indirectly.

We added the summary of the two analyses to the main text (Lines 107-113 and Lines 150-155). Details of the two analyses were added to Supplementary Discussion 2.

We think these results can answer to the three questions the reviewer raised. Our results suggest that Southern Africa and Australia are the main sources of AN iron in the Southern Hemisphere and that industry and transport are potential large emission sources of magnetite (answer to Q3). Since our simulations were evaluated for both industry-source dominant (East Asia) and transport-source dominant (Europe) regions (Supplementary Fig. S2), the extension of magnetite data to other regions where industry and transport sectors are dominant, such as Southern Africa and Australia, is therefore a more reasonable assumption than applying the magnetite results to the Northern Hemisphere only (answer to Q1 and Q2).

Based on these results, we also added a paragraph for highlighting important observations in the future to the main text (Lines 277-295). This description is an

important guideline for researchers related to iron observations and a main conclusion and interpretation of this study.

====

Lines 107-113: “Observations of magnetite concentrations (or magnetite/BC) are missing over North America and the Southern Hemisphere. In addition, emission sources of magnetite are currently not known well. However, the estimation of possible emission sources of magnetite from BC emission inventories and the underestimation of combustion iron around Southern Africa and Australia support the validity of extending the magnetite results in East Asia to the Southern Hemisphere where we focus on in soluble iron estimates (Supplementary Discussion 2).”

Lines 150-155: “Since combustion iron is dominant and iron/BC ratios are underestimated around Southern Africa and Australia in our simulations, combustion iron emissions (from AN or BB sources, or a combination of both) are likely underestimated over these regions (Supplementary Discussion 2.2). This suggests that the underestimation of AN iron emissions is a main source of the underestimation of total iron around Southern Africa and Australia (Supplementary Discussion 2.2).”

Lines 277-295: “Since many factors contribute to the large uncertainties in the estimation of magnetite DRF and soluble iron deposition flux to the oceans, diverse observational data will therefore be required in order to constrain them (Supplementary Discussion 1). First, simultaneous measurements of magnetite and total iron are needed to reduce the uncertainty in the fraction of magnetite to AN iron. Second, more field and laboratory experiments are needed to better understand processes which control solubility of iron. Third, reducing the uncertainties in the imaginary part of the refractive index for magnetite and BrC is necessary to reduce radiative forcing uncertainty. Fourth, due to the logistical constraints of ship-borne measurements, most available iron data have a limited time coverage over oceans. Longer-term observations are needed to better estimate iron amounts, especially within southern oceans. Fifth, due to the limitation of the instrument (Methods), magnetite particles greater than 3 μm are not considered in our new estimates and further measurements of coarse magnetite particles are necessary. Finally, more data on magnetite and combustion iron are needed, especially within Southern Africa and Australia and their outflow regions. Data collection should ideally be focused on constraining uncertainties in soluble iron deposition flux to southern oceans with respect to its source contributions (AN or BB; Supplementary Discussion 2.2) in order to quantify the impact of AN activity on biogeochemistry and CO_2 uptake over the southern oceans.”

Supplementary Discussion 2:

2. Extension of magnetite results in East Asia to the Southern Hemisphere

Since observations of magnetite concentrations (or magnetite/BC) are missing in the Southern Hemisphere, it is not currently possible to show direct evidence as to whether we can extend the magnetite results in East Asia to the Southern Hemisphere (Southern Africa and Australia). However, we show the validity of the extension to the Southern Hemisphere indirectly from the following two analyses using emission inventories and total iron observations.

2.1. Potential emission sources of magnetite

First, we examine potential emission sources of magnetite over East Asia, Europe, Southern Africa, and Australia by using the BC emission inventory used in this study (Lamarque *et al.*, 2010). The 4 results below (a-d) suggest that the extrapolation of the magnetite results in East Asia and Europe (Supplementary Fig. S2) to Southern Africa and Australia is a more reasonable assumption than applying the magnetite results to the Northern Hemisphere only.

- a. Magnetite and BC observations show that these two species have similar temporal variations in both East Asia (Moteki *et al.*, 2017) and Europe (Supplementary Fig. S13 at Zeppelin). Though emission sources of magnetite are not known well, this result suggests that magnetite and BC have similar emission sources (from fossil fuel combustion).
- b. Observed magnetite/BC ratios in East Asia and Europe are reproduced better in the NEW simulation (Supplementary Fig. S2). This result suggests that magnetite emissions used in the NEW simulation are reasonable over East Asia and Europe.
- c. BC from fossil fuel combustion is emitted mainly from the industry and transport sectors (Lamarque *et al.*, 2010). In East Asia (100°-140°E and 20°-45°N), industry sector is dominant (industry and transport sectors account for 78% and 18% of total BC emissions from fossil fuel combustion, respectively). In Europe (0°-60°E and 35°-70°N), transport sector is dominant (industry and transport sectors account for 31% and 62% of total BC emissions from fossil fuel combustion, respectively). These results suggest that industry and transport sectors may be main sources of magnetite and that the new magnetite emissions used in the NEW simulation are suitable for both the industry and transport sectors.
- d. BC emissions from industry and transport sectors are comparable over Southern Africa (0°-60°E and 0°-40°S, 44% from industry and 42% from transport) and Australia (110°-180°E and 10°-50°S, 37% from industry and 46% from transport). The two sectors account for 87% and 84% of total BC emissions from fossil fuel combustion over Southern Africa and Australia, respectively. These results suggest the extension of the results in East Asia (industry sector dominant) and Europe (transport sector dominant) to Southern Africa and Australia is a reasonable assumption.

2.2. Underestimation of combustion iron over Southern Africa and Australia

Second, we compare model simulations with BC and total iron observations in the Southern Hemisphere. The 5 results below (e-i) suggest that combustion iron (AN or BB sources or both) over Southern Africa and Australia and their marine outflow regions is underestimated in the model simulations and that the agreement with the iron observations is improved by extending the magnetite results in East Asia to the Southern Hemisphere (NEW simulation).

- e. BC concentrations near the surface are well simulated (or slightly overestimated) in the Southern Hemisphere in our model, as shown by Figs 9g-9h of Matsui and Mahowald (2017). This result suggests that BC emissions and concentrations are valid in the southern hemisphere.
- f. In contrast, surface iron concentrations and its deposition flux are underestimated around Southern Africa and Australia (Supplementary Figs S6-S7). The iron/BC

emissions ratio is therefore likely underestimated within these Southern Hemisphere regions.

- g. The contribution of mineral dust to total iron concentrations is low (<50%) around Southern Africa and Australia (Supplementary Figs S7c-S7d). This result suggests that combustion iron emissions (AN and/or BB) are underestimated within these regions.
- h. Simulated iron concentrations are comparable to observations around Amazon where the BB contribution is high (Supplementary Fig. S7e), suggesting the following two possibilities. One is the underestimation of the iron/BC ratio in AN emissions within Southern Africa and Australia. The other is the underestimation of the iron/BC ratio in BB emissions within Southern Africa and Australia because the iron/BC ratio within these regions may not be the same as (higher than) that over Amazon (the iron/BC ratio in BB emissions is assumed to be the same over Amazon and over Southern Africa and Australia in our simulations). One of these two possibilities or both contribute to the underestimation of combustion iron around Southern Africa and Australia.
- i. The NEW simulation has better agreement with the observations of iron concentrations and deposition flux than the BASE simulation around Southern Africa and Australia (Supplementary Fig. S7e). This suggests AN iron emissions in the NEW simulation are more realistic than those in the BASE simulation.

More magnetite and combustion iron observational data are needed in the Southern Hemisphere, particularly downwind of Southern Africa and Australia. Furthermore, attribution of combustion iron to AN and BB sources is needed to quantify the relative contribution of these sources. However, the results shown above suggest that the underestimation of AN iron emissions is a main source of the underestimation of combustion iron concentrations around Southern Africa and Australia.

Supplementary Figure S6. Deposition flux of total iron. Observed and simulated total iron deposition flux for the BASE (blue) and NEW (red) simulations. Observed data are derived from Mahowald *et al.* (2009).

Supplementary Figure S7e. The ratio between simulated and observed total iron concentrations in the Southern Hemisphere (5-year mean).

Supplementary Figure S13. Scatterplots between magnetite and BC concentrations at Zeppelin. a-b, Scatterplots of hourly observed data for (a) number and (b) mass concentrations.

Reviewer's comment:

On page 20, “intensive surface measurements at Zeppelin” are mentioned. These should be introduced more clearly, and that location placed in better context. The only information that is included is the location (79N) and duration of sampling (1 month). The supplementary information does not elaborate on the source of this data. How do these measurements fit in with the conclusions of the study? Are European emissions any more relevant to the Southern Hemisphere than Asian aerosols?

Response:

We added more description on intensive surface measurements at Zeppelin to Methods (Lines 491-499). As described in the response to the first comment, the results in Europe can be used to support the extension to other regions where transport sector is the main source of fossil fuel combustion. The results in Europe are therefore useful to extend the results in East Asia and Europe to the Southern Hemisphere (Southern Africa and Australia).

====
Lines 491-499: “Measurements at Zeppelin (78.9°N, 11.9°E, 474 m above sea level) were conducted in March (28 February to 29 March) 2017. The observation site is representative of European Arctic background and can be influenced by pollutants transported from Europe and Russia (Beine et al., 1996; Ström et al., 2003). The observed number and mass concentrations of magnetite correlated with those of BC, with the square of the correlation coefficient (R^2) of 0.90 and 0.50, respectively (Supplementary Fig. S13). These results suggest that emission sources of the observed magnetite and BC were spatially similar, as is also seen in the Asian continental outflow (Moteki et al., 2017) and urban sites in Japan (Ohata et al., 2018).”
====

Reviewer’s comment:

It may be that the authors have high confidence in the model results, for the Southern Hemisphere in particular, but the reasoning supporting that confidence is readily clear to me. I am not a modeler and could be missing an important aspect of that portion of the work, but then other readers may reach the same interpretations as I have. If this manuscript is revised, I encourage the authors to make the connections between the East Asian source data and the global outputs clearer.

Response:

We conducted the two analyses described in the response to the first comment (Supplementary Discussion 2) and suggested the validity of extending the East Asian and European data to the Southern Hemisphere. In addition to these analyses, we compared model simulations with total iron observations at Gosan in East Asia and showed the validity of our new iron emissions used in the NEW simulation. The NEW simulation could capture observed total iron concentrations at Gosan more realistically than the BASE simulation and the simulation in Luo et al. (2008) (Supplementary Fig. S4). We added these results to the main text (Lines 123-131). We strengthened the results and interpretations obtained in this study by these analyses.

====
Lines 123-131: “Comparisons of model simulations with total iron (AN+BB+Dust) observations at Gosan in East Asia¹ show that the iron emissions used in the NEW simulation are more consistent with the measurements than those used in the BASE simulation and Luo et al.¹. The NEW simulation (annual mean total iron concentrations are $0.66 \mu\text{g m}^{-3}$) has a good agreement with the total iron observations¹ ($0.63 \mu\text{g m}^{-3}$), whereas simulations in the BASE case ($0.36 \mu\text{g m}^{-3}$) and Luo et al.¹ ($0.21 \mu\text{g m}^{-3}$) underestimate iron concentrations by a factor of 2 or 3 (Supplementary Fig. S4). Supplementary Fig. S4 also shows that the NEW case can simulate the observed iron concentrations in both summer and winter (when the contribution from Dust iron is limited, i.e. AN iron is dominant).”

Supplementary Figure S4. Monthly variations of total iron concentrations at Gosan, East Asia. Observed total iron data (black) from Fig. 6 of Luo *et al.* (2008). Simulated total iron results are shown for the BASE (blue) and NEW (red) simulations (5-year mean). Iron concentrations from dust sources only is shown for the BASE simulation (orange, 5-year mean).

====

REVIEWERS' COMMENTS:

Reviewer #1 (Remarks to the Author):

The authors have revised the manuscript to include supporting materials on applying the magnetite Fe/BC ratios observed near East Asia and Zeppelin to anthropogenic emissions of Fe and BC in other regions of the world, and to include additional discussions on measurement and model uncertainties. I recommend its publication in Nature Communications.

Two minor edits:

- (1) Page 5, line 113: delete "in" after "focus on: .
- (2) Page 18, line 374: "a factor of 2" or "a factors of 5"?

Two relevant references could be cited:

- (1) Y. Takahashi, M. Higashi, T. Furukawa, and S. Mitsunobu: Change of iron species and iron solubility in Asian dust during the long-range transport from western China to Japan, *Atmos. Chem. Phys.*, 1, 11237-11252, 2011.
- (2) Y. Takahashi, T. Furukawa, Y. Kanai, M. Uematsu, G. Zheng, and M. A. Marcus: Seasonal changes in Fe species and soluble Fe concentration in the atmosphere in the Northwest Pacific region based on the analysis of aerosols collected in Tsukuba, Japan, *Atmos. Chem. Phys.*, 13, 7695-7710, 2013.

Although their estimate of mineral Fe composition is not unambiguous.

Reviewer #2 (Remarks to the Author):

The authors have responded to my previous comments to my satisfaction and this manuscript should be published in Nature Communications.
I think that this work will stimulate future measurements and analysis and progress in its topic.

Reviewer #3 (Remarks to the Author):

Left comments for the editor only.

Response to reviewer #1

NCOMMS-17-28537A: “Anthropogenic combustion iron as a complex climate forcer”
by H. Matsui et al.

We thank the reviewer very much for reading the paper carefully and giving us valuable comments. We revised the paper by taking into account the reviewer’s comments. Responses to individual comments and suggestions are given below.

Reviewer’s comment:

(1) Page 5, line 113: delete "in" after "focus on:.

Response:

We have deleted “in” from this sentence (Line 116).

Reviewer’s comment:

(2) Page 18, line 374: "a factor of 2" or "a factors of 5"?

Response:

We have revised the sentence as follows: “In the NEW simulations, the emission flux of AN magnetite was assumed to be double the AN iron emission flux in Luo et al. (BASE simulation)” (Lines 371-372).

Reviewer’s comment:

Two relevant references could be cited:

(1) Y. Takahashi, M. Higashi, T. Furukawa, and S. Mitsunobu: Change of iron species and iron solubility in Asian dust during the long-range transport from western China to Japan, *Atmos. Chem. Phys.*, 1, 11237-11252, 2011.

(2) Y. Takahashi, T. Furukawa, Y. Kanai, M. Uematsu, G. Zheng, and M. A. Marcus: Seasonal changes in Fe species and soluble Fe concentration in the atmosphere in the Northwest Pacific region based on the analysis of aerosols collected in Tsukuba, Japan, *Atmos. Chem. Phys.*, 13, 7695-7710, 2013.

Response:

We cited these papers in our revised manuscript (papers 13 and 39 in the reference list).